# Is Türkiye's on-the-job training program sustainable? A qualitative evaluation from the firms' perspective

**Mustafa Koçancı¹, Mete Kaan Namal², Beyhan Aksoy📙²\*, Ufuk Bingöl³, Zihni Onur Eren⁴**

**1** Faculty of Economics, Administrative and Social Sciences, Alanya Alaaddin Keykubat University, Antalya, Türkiye, **2** Faculty of Economics and Administrative Sciences, Akdeniz University, Antalya, Türkiye, **3** Manyas Vocational School, Bandırma Onyedi Eylül University, Balıkesir, Türkiye, **4** Akdeniz University, Antalya, Türkiye

\* beyhanaksoy@akdeniz.edu.tr

## Abstract

Active labor market policies play a vital role in achieving sustainable development goals. This study examines the effectiveness of the on-the-job training (OJT) program, a key active employment policy implemented by the Turkish Employment Agency (TEA), within the context of Antalya province, Türkiye. The research aims to explore employers' perceptions of the program, its effectiveness, and its contribution to sustainable development. Data were collected through semi structured interviews with managers responsible for OJT in various firms and analyzed thematically using computer assisted tools. The findings show that although the program contributes to the increase of human capital by creating employment and developing skills, it falls short of its goals due to labor segmentation issues, operational inefficiencies, and its inability to close implementation gaps, revealing that the program faces a number of structural challenges. The study concludes that although OJT holds significant potential for promoting sustainable employment, further improvements are necessary to enhance its impact.

## 1. Introduction

In order to protect social welfare, ensure economic stability, and reduce unemployment, governments pursue a variety of policies and programs for the labor market. However, the rise in unemployment, especially in developed economies in the 1980s, led governments to place greater emphasis on developing and coordinating labor market policies to provide social protection [1].

Labor market policies employ two approaches to combat unemployment: active and passive labor market policies. Passive labor market policies (PLMPs) were first implemented in Europe. Its most general scope is to provide financial support to individuals during periods of unemployment. These policies encompass unemployment

**Data availability statement:** All relevant data are within the manuscript and its Supporting Information files.

**Funding:** The author(s) received no specific funding for this work.

**Competing interests:** The authors have declared that no competing interests exist.

insurance, unemployment benefits, etc. With the neoliberal transformation since the 1980s, policies to combat unemployment have also diversified. Active labor market policies (ALMPs implemented since the early 1990s have been more concerned with bringing people back to work by providing them with the necessary knowledge, experience and equipment than supporting them financially while they are unemployed.

ALMPs are categorized into two main categories. The first group aims to improve the individual matching process for the unemployed to find and settle in jobs. The second group of ALMPs aims to enhance individual productivity. In short, the former constitutes job search assistance programs, while the latter include training programs, workfare benefits, and incentivized employment [2]. In this context, policies such as vocational training, job search assistance, and on-the-job training are examples of active labor market policies. Despite the widespread belief that public spending should shift from passive policies to active interventions, countries implementing this strategy have not automatically improved their labor market performance. This suggests that active and passive policies should be viewed as two key components of a broader social protection system [1].

ALMPs aim to bridge job seekers and employers through employment counseling and labor market information services [3–6]. These policies encompass a range of interventions such as wage and employment subsidies, self-employment support, public job creation, job matching, and vocational training initiatives [7]. Despite their comprehensive scope, the effectiveness of these measures in sustainably increasing employment remains debated [8].

Vocational training programs aim to activate and reintegrate unemployed individuals,particularly the long-term unemployed and youth, by aligning their competencies with the evolving needs of production and service sectors [9]. When structured according to sectoral analysis, these programs can provide companies with qualified labor, yet their success may increase further when they focus on disadvantaged social groups rather than the broader population [9,10]. International examples, such as the U.S. Job Corps Program, highlight how such initiatives can promote socio-economic integration, reduce crime, and raise income levels among young participants [11]. Overall, vocational training represents both a social investment and a policy instrument central to the logic of active labor market strategies, aiming not only to reduce unemployment but also to strengthen social inclusion through skill development. Given their multidimensional scope, active labor policies, particularly vocational training initiatives, are also increasingly linked to broader socio-economic objectives such as sustainable development and inclusive growth.

Labor market characteristics and government interventions are key to sustainable development, as labor market policies aim to boost economic growth, reduce inequalities, enhance employment, promote social stability, and ensure shared welfare [12]. In 1987, the United Nations Brundtland Commission defined sustainability as "meeting the needs of the present without compromising the ability of future generations to meet their own needs" [13]. The Sustainable Development Goals, adopted by the United Nations [14] in 2015 and defined under 17 sub-headings, have the main objectives of ending poverty, protecting the environment, taking measures against

the climate crisis, fair sharing of wealth, and peace. Labor market policies have more explicit links with some of the UN development goals and are more likely to contribute to them.

Although the OJT programs are presented as a contributor to sustainable development, particularly through SDG 8, its sustainability remains under theorized in labor policy literature. Sustainability in this context should not only be interpreted as temporary employment generation. To clarify what is meant by "sustainability" in the context of OJT programs, sustainability is conceptualized along three interlinked dimensions: economic viability (firms' productivity returns and the conversion of placements into stable jobs), social inclusion (equitable access and outcomes across gender and other vulnerable groups), and institutional stability (continuous, adaptive public–private partnerships and implementation capacity). Sustainable employability frameworks emphasize long term workability and capability building rather than short term placement alone [15]. Dual or firm linked training arrangements that combine workplace and school based learning tend to deliver higher labor market returns highlighting the importance of institutional embedding [16]. Finally, closing the policy–practice gap is essential: bureaucratic rigidity, weak coordination, and limited local adaptability undermine the long term sustainability of OJT program outcomes [17]. These dimensions mu(st be evaluated not only by participation rates but by outcomes that contribute to decent work as defined in SDG 8.

Development is about enhancing human potential by building new structures and adapting to change, whereas human centered development emphasizes empowering individuals to achieve what they value [18]. In this context, the study assesses how employers perceive OJT programs implemented by the Turkish Employment Agency (TEA) and evaluates their functionality and contribution to sustainable development in Antalya as a tourism metropolitan city. By focusing on individual empowerment through the OJT program, the study examines its role in fostering human-centered development and broader socioeconomic growth.

The study examines the effectiveness of OJT programs by drawing on multiple theoretical perspectives that offer complementary insights into their mechanisms and outcomes. This study adopts a compact analytical framework that integrates Human Capital Theory, Labor Market Segmentation, Policy–Implementation Gaps, and research on public–private coordination in active labor market policies (ALMPs). Human Capital Theory elucidates how OJT program enhances productivity and employability through skill development, while Labor Market Segmentation highlights that such benefits are not distributed evenly across different strata of the labor market, where access to opportunities and career progression may be structurally constrained. The policy–implementation gap perspective provides a valuable lens for understanding the discrepancies between program design and real world application, especially in contexts where institutional capacity and administrative coherence are limited. Finally, insights from studies on collaborative governance clarify how the interaction between state institutions and firms influences program participation, training content, and the translation of temporary placements into stable employment Rather than employing these frameworks merely in a descriptive manner, the analysis uses them to interpret how incentive structures, selection practices, and employers' emphasis on behavioral attributes may either perpetuate or alleviate existing inequalities within the labor market.

While previous studies have examined OJT from a macroeconomic or policy perspective, few have focused on employer centered qualitative assessments, particularly in the context of regionally distinct labor markets. By exploring TEA administered OJT programs in Antalya, a tourism driven, seasonally dynamic labor market, this study offers novel insights into how ALMPs operate at the intersection of firm strategy, state policy, and labor market fluctuations. This structural instability inherently challenges the ALMPs' goal of employment stabilization. By investigating the OJT in this context, the study aims to provide critical insights into the program's effectiveness not only in Turkey but also within other economies facing high seasonality and employment instability (e.g., Mediterranean and developing tourism-dependent regions). Furthermore, by analyzing firm-level experiences and challenges, the study contributes to closing the empirical gap in the literature regarding the sustainability, inclusivity, and effectiveness of publicly funded OJT programs in Türkiye. This approach allows for critical engagement with the literature, moving beyond descriptive summaries to offer practical and policy-relevant conclusions.

 

## 2. Literature review: Skill development, implementation challenges and inequalities

Unemployment policies today are largely categorized into two main approaches: demand side macro policies and supply side micro policies. Many countries, including Türkiye, develop their strategies to address unemployment based on these frameworks. However, the sustainability of unemployment policies is crucial to ensure long term economic stability, social inclusion, and the continuous development of the labor market. Accordingly, the study separately examines how active labor market policies contribute to skill development processes, particularly through vocational training and OJT, the institutional and structural challenges encountered during the implementation phase, and the impact of these policies on inclusivity and social inequalities. The following three sections explore the multi-layered nature of these policies in greater detail.

### 2.1. OJT programs and skill development

Training based ALMPs aim to equip unemployed individuals with the skills required to re enter the labor market. Vocational training and OJT programs are among the most widely applied instruments to this end [19,20].These programs are designed to enhance employability through practical, workplace based experience [2]. OJT programs play a critical role not only in terms of short term employment outcomes but also in the development of human capital and the reduction of income inequalities. Researches show that OJT, which encompasses both formal training provided by employers and informal learning processes, is essential for skill acquisition and for maintaining competitiveness in rapidly changing markets [21–24]. In high income countries in particular, a greater proportion of workers receive employer sponsored training, which contributes significantly to global disparities in wage growth and income levels [22,25]. Moreover, investment in OJT is not merely a response to immediate labor market demands but also a strategic approach aimed at enhancing long term productivity and innovation within firms [26].

OJT programs enhance human capital by improving vocational competencies and strengthening employees' abilities to cope with challenges in the workplace. The demand for training and intervention based programs has been increasing, as such programs support occupational effectiveness and engagement [27]. In terms of their aim to develop human capital, OJT programs hold significant potential for promoting sustainable development. Al Frijat and Elamer (2024) [28] argue that human capital supports institutional sustainability in developing countries; Kichurchak (2024) highlights the need for public investment in education due to its links with employment and resource efficiency [29]; and Serikkyzy et al. (2024) [30] demonstrate that investments in education and health promote both economic and environmental sustainability.

In Türkiye, vocational training and OJT programs led by the TEA form a significant pillar of ALMPs [31]. Examining publicly funded OJT programs in Turkey through a human capital lens is crucial to understanding how effectively these initiatives enhance individuals' knowledge, skills, and competencies. Such an approach not only helps align workforce development with labor market needs but also ensures the efficient use of public resources. Furthermore, evaluating OJT programs from a human capital perspective sheds light on their long term contributions to sustainable development by linking individual skill formation with broader economic and social welfare goals [32,33]. The goal is to bridge the skill gap in the labor market and provide job specific competencies. However, studies also point out limitations in achieving targeted outcomes, particularly when training is not tailored to labor market needs or when participants lack long term integration opportunities [2].

### 2.2. Firm level implementation and experiences

Policies often fail due to the complexities inherent in the implementation process. The gap between policy design and practice is not merely a technical deficiency but a structural issue shaped by multidimensional factors such as policy frameworks, institutional arrangements, and political context [34,35]. The Implementation gap perspective offers a useful framework to explain the divergence between intended policy outcomes and real world results, emphasizing that policy

failures often emerge during the implementation phase [17]. Key contributors to this gap include institutional misalignments, limited knowledge and capacity among implementers, and weak communication among stakeholders [36]. Therefore, policy design must take into account both the local context and the specific needs of the target groups.

Hupe et al. (2014) argue that the implementation gap cannot be fully understood without considering the multidimensional nature of the implementation process [37]. Hudson et al. (2019) emphasize that overly optimistic expectations, fragmented governance structures, insufficient collaboration in policymaking, and short term approaches compounded by political uncertainty all contribute to policy failure [17]. To overcome these challenges, the authors propose several tactics including fostering collaborative policy design, developing robust implementation statements, utilizing evidence based practices, establishing clear communication processes, and recruiting experienced "implementation brokers [38]."

Successful implementation processes require a joint analysis of environmental, institutional, and individual level factors [39,40]. Ambiguity in policy objectives can lead to informal discretion among managers, resulting in inconsistent implementation and failure to achieve intended outcomes [41]. This discretion is often shaped by broader structural conditions that influence both managerial decision making and participants' opportunities. Thomsen (2009) found that structural barriers account for a significant portion of the variation in job finding outcomes [42]. He therefore argues that ALMP programs should not focus solely on skills training, but rather be restructured through holistic approaches that address broader structural challenges.

This multi layered divergence between policy and practice is directly reflected in the experiences of field level actors and institutional implementation processes. In the context of OJT programs, the implementation gap perspective provides a valuable analytical bridge between policy intent and organizational practice, highlighting how misalignments between the two undermine program effectiveness. As noted by Martin et al. (2025) [43], the policy–practice gap represents a dual challenge: organizations must both adapt their practices to policy demands and recalibrate policies to fit practical realities. Applying this perspective to OJT programs allows for capturing how limited flexibility, weak coordination, and inadequate local adaptation hinder the translation of policy goals into sustainable employment outcomes. In this context, the specific challenges encountered by firms during the implementation of OJT programs and the dynamics shaping this process are of critical importance for understanding how the implementation gap manifests in practice. Despite the growing application of OJT programs, the literature capturing the employer's perspective and experience remains limited. Much of the empirical work focuses on quantitative outputs such as employment rates or income changes, often overlooking qualitative experiences and organizational dynamics [44,45]. Chandler (2000) and Hsieh et al. (2009) argue that OJT provides firms with flexible mechanisms to address evolving skill requirements and operational needs [46,47]. TEA's OJT programs, by offering financial incentives and social security coverage, aim to reduce the cost burden on firms and promote early employment pathways. However, Timsal et al. (2016) highlight that without systematic planning, firms may underutilize OJT or fail to meet trainee expectations, reducing program effectiveness [48]. Moreover, the literature rarely discusses the potential disincentives and administrative burdens firms encounter. Employers may be reluctant to hire trainees full time after the OJT program period due to bureaucratic complexity or concerns about social security liabilities [49].

### 2.3. Gender, inclusivity and sustainability

ALMPs are closely tied to the United Nations Sustainable Development Goals (SDGs), particularly those concerning gender equality (SDG 5), decent work and economic growth (SDG 8), and reduced inequalities (SDG 10). However, while policy rhetoric often emphasizes these links, implementation tends to lag behind [13]. Women and youth are frequently cited as target groups for activation policies, yet the actual inclusivity of OJT programs is often questioned. For instance, Jianu et al. (2021) note persistent gender disparities in access to vocational training across the EU [12].

The gap between program design and real world outcomes raises critical questions about equity and sustainability. While TEA's programs are comprehensive on paper, their practical effects on gender equality and inclusive employment

are limited without stronger enforcement mechanisms or incentive structures. This observation echoes critiques about the need to integrate a rights based approach into ALMPs [18].

OJT programs, aim to enhance individuals' employability, facilitate their access to employment, and promote the creation of new jobs. However, the effectiveness of such programs is influenced by structural features of the labor market that shape how individuals can access and benefit from employment opportunities [50,51]. In this regard, Labor Market Segmentation Theory highlights the fact that the labor market is divided into distinct segments and that transitions between these segments can be quite limited [52]. This perspective generally divides the labor market into two main sections: the primary sector, characterized by high wages, job stability, and good working conditions, and the secondary sector, defined by low wages, insecurity, and poor working conditions [53,54]. This division is not limited to wage levels alone but also entails significant disparities in terms of employment conditions, opportunities for promotion, and job security [55,56]. Institutional and societal factors, such as trade unions, employer practices, and immigration status, play a decisive role in the emergence and persistence of segmentation [53,57,58]. Although the form of segmentation varies across countries and over time, these structural divisions remain persistent. Therefore, taking segments into account can highlight the constraints on labor mobility and the unequal capacities of individuals to translate their education into sustainable employment outcomes. This, in turn, allows for a more critical analysis of OJT programs [59,60].

The existence of structural inequalities in the labor market suggests that the impacts of interventions like OJT programs may vary significantly across different segments. Hence, policies designed without considering this structural fragmentation may fail to reduce inequalities and may instead reproduce existing divisions [51,61,62]. Since labor market segmentation substantially shapes preferences for redistribution, policy analysis must incorporate an understanding of these labor market dynamics [63]. Grimshaw et al. (2017), when discussing the disadvantaged position of certain groups, particularly women, in the labor market, emphasize that segmentation is not solely the result of individual deficits but is also shaped by demand side employer preferences and institutional strategies [64]. The concentration of women in low paid and precarious jobs is linked to the entwinement of gender roles and care responsibilities with institutional arrangements, revealing the inadequacy of individualised employability focused policies.

ALMPs should not only aim at skills enhancement but also adopt regulatory and gender sensitive approaches that address structural inequalities and discriminatory employer practices [64]. Labor market segmentation theory provides a valuable analytical lens for examining how OJT programs operate across differentiated segments of the labor market. From this perspective, OJT programs do not function in a uniform environment but rather within a structure divided into primary and secondary segments, each characterized by distinct employment conditions, mobility opportunities, and rewards. As Eichhorst et al. (2015) [16] note, vocational systems that integrate workplace and school based training, such as dual apprenticeship models, yield higher returns in job entry and wage growth than purely school based systems. This suggests that OJT programs effectiveness depends on its institutional embedding: when linked to structured learning and employer commitment, it can bridge labor market segments; when confined to low skill or precarious contexts, it risks reinforcing segmentation rather than reducing it. Therefore, in the evaluation of programs such as OJT, it is essential to assess not only employment rates but also the quality and segmentation of employment, including the sector in which employment occurs and the conditions under which it is sustained.

Despite the extensive literature on active labor market policies and on-the-job training programs, existing studies predominantly rely on macro-level evaluations and quantitative indicators such as employment rates, program participation, or income effects. Comparatively fewer studies examine OJT programs from the perspective of employers through qualitative designs that capture firm-level implementation processes and organizational decision-making dynamics. Moreover, the sustainability of OJT programs is often implicitly assumed rather than explicitly theorized, with limited attention paid to the interrelated economic, social, and institutional dimensions that shape long-term outcomes. This gap is particularly evident in regionally specific labor markets characterized by seasonality and volatility, such as tourism-oriented economies. Addressing this gap requires an employer-centered qualitative analysis that links implementation experiences to questions

of sustainability, inclusivity, and program effectiveness. Within this scope, the present study seeks to address the following research questions:

Q1: How do firms recruit participants for the OJT program, and what criteria and methods do they use?

Q2: What challenges do firms face in recruiting and retaining qualified labor under the OJT program framework?

Q3: How do firms experience and perceive the training, auditing, and implementation processes of the OJT program?

Q4: What are firms' assessments of the outcomes of the OJT program in terms of trainee performance, satisfaction, and employment impact?

## 3. Materials and methods

### 3.1. Qualitative research design

This study focuses on the firm level functioning of OJT programs, examining in detail how these programs operate based on employers' experiences and perceptions of the implementation process. In this context, a deeper understanding of the potential of OJT programs to support sustainable employment policies requires an indepth exploration of employers' attitudes toward the program, the challenges they encounter, and the outcomes they report.

This study employed an interpretive descriptive case study design and computer assisted qualitative data analysis methods to explore the perspectives of employers participating in Türkiye's OJT program. The sample included 35 firms in Antalya province, selected through purposive sampling with maximum variation to ensure sectoral diversity, including 29 in the service sector (primarily tourism), 4 in the industrial sector, and 2 in agriculture, reflecting the regional and national distribution of OJT programs. Firms were selected using purposive random sampling, and semi-structured interviews were conducted with managers responsible for OJT. This selection criterion was chosen to ensure that respondents possessed indepth knowledge of OJT implementation, training, and supervision processes. With participants' consent, interviews were recorded and analyzed according to Tong et al. (2007) criteria [65]. A hybrid coding approach was used, combining inductive thematic analysis and grounded theory coding procedures. Saturation was assessed iteratively; data collection continued until no new thematic categories emerged from subsequent interviews, following Glaser and Strauss's theoretical saturation principle.

While the initial program evaluation was predominantly a descriptive analysis, this article re-examines the topic through a distinct theoretical framework and a deeper analytical approach by enhancing the qualitative dataset (N = 35). The current study incorporates novel coding and thematic analyses—absent from the original thesis—specifically to investigate the 'sustainability' dimensions of the program (economic, social, and institutional). Thus, this paper aims to offer an original contribution to the 'implementation gap' debates in the literature. Consequently, this study offers an original re-interpretation of the raw data through a new research question and a distinct theoretical lens. In this context, the study was conducted with the approval of the Bandırma Onyedi Eylül University Social and Humanities Ethics Committee (Protocol No: 2024−7, dated 05/09/2024).

### 3.2. Data collection and analysis

Semi-structured interviews were conducted with the representative of the firms using a flexible protocol that allowed for probing of emerging themes. Interviews averaged 45–60 minutes in duration and were audio recorded with participants' informed consent. All interviews were transcribed verbatim and anonymized to protect confidentiality. The interviews were conducted in Turkish and translated into English by bilingual researchers for coding purposes, with cross validation for meaning retention. Thematic analysis of the employer centered OJT program impact focused on the experiences, observations, and expectations of business representatives, utilizing a descriptive case study design based on Creswell and Creswell's phenomenological research framework [66]. Grounded theory methodology was also employed to derive theories from participant data rather than predefined categories. This iterative process, known as "constant comparative

analysis" [67], allowed the simultaneous collection and analysis of data, fostering the development of new concepts. This approach allowed for balancing data-driven theme generation with the theoretical sensitivity needed to interpret employer perspectives within the broader context of active labor market policy. To ensure coding consistency, two authors independently performed line-by-line open coding on a subset of transcripts, achieving consensus through team discussions. Through iterative team discussions, initial codes were compared, clustered, and refined into a shared codebook. Axial coding was applied to establish relationships among categories (e.g., motivations, perceptions, implementation challenges). Codes were grouped into higher-order themes and sub-themes, such as "Vacancy and Employment," "OJT Practices," and "Program Outcomes.". Theoretical memos were used throughout to track emergent concepts, following grounded theory procedures. In addition to basic coding functions, quantitative evaluations were tabulated using NVIVO QDA software, whose data mining analytics (matrix coding queries, word frequency tools, and cluster analysis visualizations) enhanced the study's reliability [68].

While the main analysis is interpretive and thematic, NVivo's cluster analysis was used as an exploratory visualization tool to identify frequently co-occurring codes. This approach helped to confirm the internal coherence of the themes rather than to generate quantitative findings. Specifically, the clusters highlighted pairs such as discipline–motivation and cohesion–personal care, which had already emerged inductively during coding. Furthermore, this technique enabled the mapping of how specific linguistic patterns and frequently repeated phrases systematically converged around the core thematic codes, providing structural evidence for the interpretive findings. These visual outputs therefore served to triangulate and reinforce the credibility of the qualitative themes rather than replace interpretive reasoning. To ensure transparency, a brief description of the procedure and the resulting clusters is included in Appendix, and the main analysis continues to rely on narrative-based interpretation.

Although primarily adopting an interpretive thematic approach, a supplementary cluster analysis using NVivo's co-occurrence matrix tools was included to visually explore overlapping patterns in firms' responses. However, such quantitative mapping may risk oversimplifying individualized and context-specific narratives, as cluster analysis can inadvertently obscure the nuanced and personal meanings that are foundational to qualitative inquiry. Accordingly, the primary analytical focus remains firmly rooted in narrative-based interpretation, and these quantitative outputs are treated as exploratory visuals rather than conclusive findings. Within the scope of the research questions, the main problem experienced by firms in employment concerns the supply of qualified personnel, particularly in relation to regional characteristics and peak-season dynamics.

In addition, the raw dataset was anonymized and uploaded to the journal system. Researchers interested in this topic may access the dataset, and the NVivo file can be made available upon justified request to the corresponding author.

### 3.3. Validity and reliability

Three separate thematic analyses were conducted to assess the study's validity [69] using QDA software. The analyses achieved an average coding agreement of 83.66%, aligning with the thresholds established by Miles and Huberman for first and second level themes [70]. Consistent with Corbin and Strauss [71], the repeated presence of participants' explanations across the dataset confirmed the appropriateness of the thematic analysis. Reliability was ensured through completion and verification checks, comparing results across analyses. To ensure analytical rigor in terms of triangulation, reflexivity and audit trail, the team revisited transcripts to verify that constructed themes were grounded in multiple data points and that no major divergence occurred across coders. Disagreements were resolved through discussion until consensus was reached. These processes were documented and reported following the QDA consolidated criteria for qualitative research under methodological domains [65].

### 3.4. Scope and limitations

Antalya holds a significant share in OJT activities in Türkiye and considering the tourism dynamics and seasonality of tourism in this region, its large role in OJT can be understood. Although Antalya's labor market is strongly shaped by its

tourism sector, the rationale for selecting this province extends beyond sectoral specificity. Antalya is one of the leading provinces in Türkiye in terms of participation in the OJT program implemented by the TEA, encompassing not only tourism but also service, manufacturing, and agricultural sectors. This internal sectoral diversity allows for the examination of both localized and cross sectoral implementation patterns. Furthermore, Antalya's high labor turnover and pronounced seasonality create a revealing context in which systemic challenges, such as bureaucratic rigidity, skill mismatches, and employment sustainability, become more visible. As such, the region provides a critical lens through which the functionality and institutional design of the national OJT system can be examined.

Accordingly, the sample for this study was selected from representatives of businesses operating in Antalya. The analysis for Antalya also contributes to a deeper understanding of how ALMPs function in practice, particularly in the context of local and regional labor markets characterized by volatility and seasonality. Informed consent was obtained from all participants, and participant confidentiality was ensured. However, as with all qualitative research, the generalizability of the data to the population is limited. It is known that different sectors have different dynamics. With regard to public incentives, general apprenticeship training with incentives such as tax deductions for OJT programs represents a common active labor market policy across the globe. However, Turkey has been selected for the purpose of analyzing the outputs of the TEA OJT program, which is integrated with direct daily wage and insurance support, tax reduction and has been implemented in a different effective way from the traditional ALMP since 2009. Conversely, the rationale behind selecting Antalya province, which exhibits a considerable labor turnover and potential for seasonal employment, is that other sectors can also derive benefit from OJT, in addition to the tourism sector. The limitations of this study also include the lack of financial support and the regional scope of the sample.

Although the importance of including employee and trainee perspectives is acknowledged, the scope of this study is deliberately limited to employer perceptions for two reasons: first, to examine program implementation from the perspective of operational actors responsible for training and recruitment decisions; and second, to evaluate how firm-specific experiences shape perceived program effectiveness and feasibility. The employer point of view provides insightful information about how companies perceive and make use of public training incentives and if mechanisms of that sort respond to labor market requirements.

## 4. Results

The findings are presented according to the four main research questions, each supported by thematic categories and illustrative excerpts from the interviews, as shown in Tables 1–3. Employment policies were categorized by firms' criteria for employees, challenges in finding qualified personnel, and methods for filling vacancies. The second part examined OJT practices, including firms' motivations for adopting OJT, training and supervision procedures, and occupational groups involved. The final section focused on firms' views on OJT outcomes, assessing satisfaction with trainee performance, comparisons between OJT and full-time employees, criteria for transitioning trainees to full-time roles, and criticisms and suggestions for program improvement.

### 4.1. Recruitment practices and challenges

Firms participating in the OJT program applied a diverse set of recruitment criteria, shaped largely by sectoral needs. Most employers emphasized prior experience, education, foreign language skills, and age as key factors.

Employment criteria vary by firm type (Fig 1). Tourism and hotel management firms, the most represented in the sample, differ from multifaceted service (MFG), textile, and health firms, which prioritize education, age, work discipline, and gender equality over experience. MFG firms in the service sector aim to train their workforce rather than hire experienced candidates. Foreign language proficiency is emphasized primarily for hotel establishments and managerial roles outside the tourism sector. Age criteria are notable in tourism, with a preference for employees aged 18–35, though some firms prioritize professional competence over age. One hotel manager remarked:

**Table 1. Theme and Coding Map for OJT Program Impact Analysis of Vacancy and Employment.**

| Theme | Categories | Codes | Number of Coding per Firm |
|---|---|---|---|
| Vacancy and Employment | Criteria | Experience | 28 |
| | | Age | 23 |
| | | Education | 23 |
| | | Foreign Language | 21 |
| | | Gender Equity | 17 |
| | | Discipline | 14 |
| | | Appearance and Self-Care | 6 |
| | | Harmony | 2 |
| | | Flexible Working Hours | 1 |
| | | Travel | 1 |
| | | Mandatory Military Service | 1 |
| | | Location | 1 |
| | Challenges | Qualification | 25 |
| | | Temporary/Seasonal Employment | 20 |
| | | Wages | 17 |
| | | Reluctance of Employees | 11 |
| | | Accommodation | 4 |
| | | Student Employee | 5 |
| | | Firm's Approach | 2 |
| | | Transportation | 2 |
| | | Urban Challenges | 2 |
| | | Voluntary Unemployment | 2 |
| | | Shifting Hours | 2 |
| | | Baby Care | 1 |
| | Qualified Workforce Employment Challenges | Seasonal Challenges | 12 |
| | | Inexperiences | 6 |
| | | Overfitting Issues | 4 |
| | | TEA Challenges and Benefits | 4 |
| | Announcement and Employment Orientations | TEA | 28 |
| | | By References | 27 |
| | | Firm's Human Resource Data Base (HRDB) | 13 |
| | | Social Networks | 10 |

*"We need young, dynamic staff who can adapt quickly, especially during the peak season. Age really matters for positions like housekeeping or waitstaff." (C11)*

Gender-based positive discrimination is generally absent, except in the tourism sector, where it is position-dependent but not a dominant factor. In this regard, the majority of interviewees explicitly asserted that gender does not play a significant role in the recruitment process. But,

*"We don't consider gender; it depends on the job. But in some positions, women just don't apply,"* stated one employer *(C04),* echoing broader cultural and structural barriers. Or, *"We don't differentiate by gender, but sometimes women don't apply for field jobs. If they do, we accept them, but the job itself is physically demanding, and they often leave".* *(C08)*

**Table 2. Theme and coding map for OJT training program impact analysis of OJT practices of firms.**

| Theme | Categories | Coding References | |
|---|---|---|---|
| | | **Codes** | **Number of Coding per Firm** |
| OJT Practice | Reasons for Applying OJT | Incentives | 31 |
| | | Socio- economic Impact | 27 |
| | | Increasing Qualification in Firm | 23 |
| | | Early Employment | 7 |
| | | Gaining Experience | 6 |
| | OJT Employment Types | Food & Beverage | 12 |
| | | Janitor | 12 |
| | | Technical | 10 |
| | | Front desk | 8 |
| | | Field Personnel | 6 |
| | | Accountant | 6 |
| | | Human Resources | 6 |
| | | Sales | 4 |
| | | Non-Qualified | 2 |
| | | Teacher | 2 |
| | | Information and communic. tech | 3 |
| | | Health | 1 |
| | | Security | 1 |
| | | Agricultural Engineer | 1 |
| | | Psychologist | 1 |
| | | Packaging | 1 |
| | | Typographer | 1 |
| | | Marketing | 1 |
| | OJT Training Procedures | Occupational Trainings | 13 |
| | | Mandatory Trainings | 17 |
| | | Encouragement | 19 |
| | | Adaptation Trainings | 11 |
| | | Skill Development | 4 |
| | TEA Auditing Issues | Constructive Approach | 5 |
| | | Procedures | 24 |
| | | Firm TEA Relations | 27 |

Employment challenges are based on various direct and indirect factors. Fig 2 shows the reported reasons for the employment challenges experienced by the firms in the research sample.

Interviews revealed a disconnect between employees' declared qualifications and their actual skills. While 78% of firms prioritize hiring through references, many also utilize job search portals. TEA, as the primary recruitment channel, poses challenges in verifying candidates' qualifications.

A recurring theme was the mismatch between stated qualifications and actual skills, especially among candidates referred through TEA:

*"Candidates often list skills they don't really have. When they start working, we realize there's a big gap between what's on paper and what they can actually do." (C03).*

**Table 3. Theme and coding map for OJT program impact analysis of OJT program outputs.**

| Theme | Categories | Coding References | |
|---|---|---|---|
| | | **Codes** | **Number of Coding per Firm** |
| OJT Outputs | Gratification from Performance of OJT | Positive | 25 |
| | | Partly Positive | 6 |
| | | Negative | 4 |
| | Comparison of OJT vs. Fulltime Contractors | No Difference | 19 |
| | | Perform More | 8 |
| | | Perform Less | 8 |
| | Transition Requirements from OJT to Full Time Contract | Harmony | 22 |
| | | Discipline | 21 |
| | | Career Goal | 18 |
| | | Talent | 17 |
| | | Self-Care | 15 |
| | | Eagerness | 8 |
| | | Reliability | 4 |
| | Impact of OJT to Unemployment | Positive | 31 |
| | | Neutral | 4 |
| | Criticism and Suggestion for Improving OJT | Social Security Coverage | 26 |
| | | Flexibility | 5 |
| | | Sanctions | 4 |
| | | Extending Expiration Dates | 6 |
| | | Bureaucracy | 5 |
| | | Non-Qualified Nominees | 1 |
| | | Specializations | 2 |
| | | Wage Adjustment | 2 |
| | | Inclusion Criteria | 2 |
| | | Occupational Health and Safety | 1 |
| | | (OHS) | 1 |
| | | Regional Assignment | 1 |
| | | Issues for OJT Employee | 1 |
| | | Certification | 1 |
| | | Announcement | 25 |

In the tourism sector, seasonal labor demands intensify these issues, forcing firms to hire students and train unskilled workers to meet peak season needs. Similar labor shortages persist in nontourism sectors during peak periods.

Seasonal work challenges arise largely from high wage expectations among skilled labor, who seek to earn enough during the season to sustain themselves year round. This forces businesses to meet these demands to attract qualified candidates. For unskilled roles, reluctance to accept minimum wage complicates recruitment. Hiring unskilled workers at or just above minimum wage creates workforce instability, with frequent turnover as employees leave for better paying opportunities.

Another reported problem in the labor supply of the research sample is the reluctance and job selection behavior of the employees. Firms reported candidates being willing to work in administrative jobs, quitting a very short time after accepting the given jobs, dismissive behavior on the part of candidates who have completed higher education regarding the

**Fig 1. OJT employment criteria by firm type.**

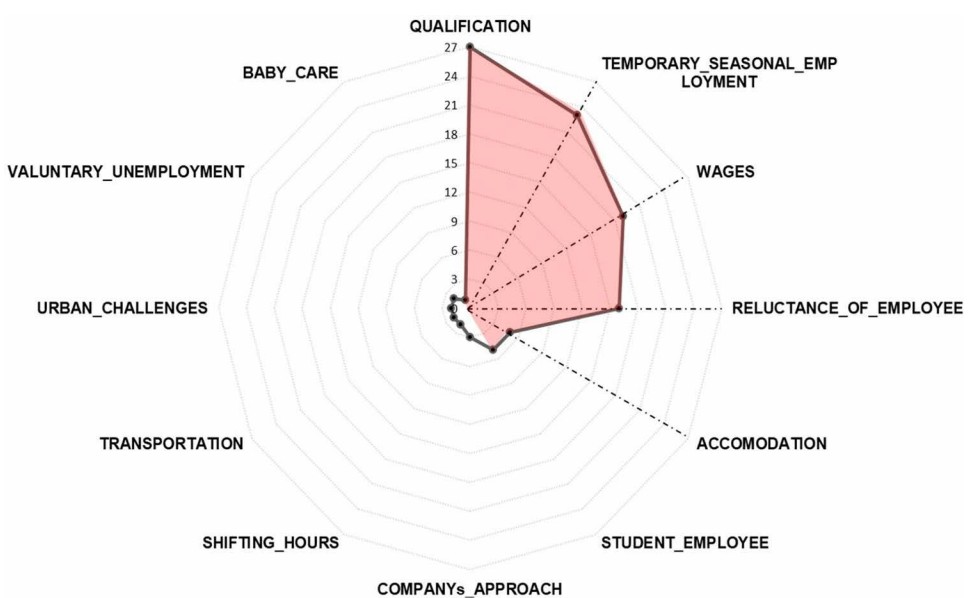

**Fig 2. Challenges in employment.**

position needed, and the perception of candidates to be recruited through TEA programs of being in a temporary employment situation.

> "They expect office roles, not field work. Even though they applied, they leave in a week when they realize what the job entails." (C16)

## 4.2. Experiences with OJT implementation

In the theme addressing firms' OJT practices, the data is categorized into four groups: Reasons for Applying OJT, OJT Employment Types, OJT Training Procedures, and TEA Auditing Issues (Table 2.). Employers overwhelmingly cited financial incentives as their primary motivation for joining the OJT program.

When the labor force requirement profile of firms benefiting from OJT in the research sample is examined, a general need for blue-collar labor is observed, although sectoral differences remain. The labor requirement profile, prepared according to the answers of the participants, is presented in Fig 3.

OJT trainees were most commonly placed in front desk services, food and beverage, janitorial roles, and technical support, with firm needs varying by sector. However, despite the program's potential, participants noted a lack of standardized training practices. Only a minority of firms offered formal orientation or adaptation sessions. One HR specialist described the challenge:

*"We train them on the job, but there's no structured curriculum. It depends on the supervisor and sometimes it's inconsistent." (C12).*

The most important reason for firms to prefer the OJT program is the financial incentives it provides. TEA's coverage of daily stipends and insurance significantly lowered hiring costs and allowed firms to take recruitment risks:

*"Without the support from TEA, we couldn't afford to train people this way. The financial aid really makes it manageable." (C07)*

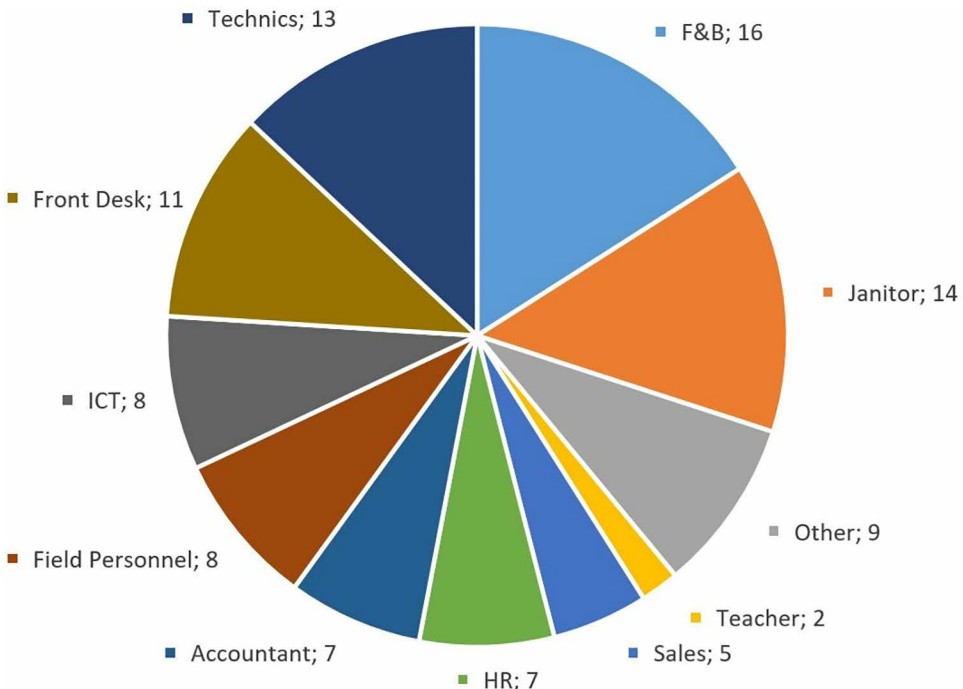

**Fig 3. Occupational groups in need.**

The firms helped by TEA and the state also reported that they provide socioeconomic benefits such as social impact and contribution to employment. While not central to the interpretive framework, the QDA clustering analysis step aimed to visualize how certain motivations, such as financial incentives, improving firm-level qualifications, and creating early employment opportunities, tended to appear together across different firm responses. These cooccurrences are summarized in Table 4.

Despite the challenge of identifying qualified labor force in the firms in the research sample, 65% of the participants indicate that they have the opportunity to train the qualified labor force they require within their own organization through OJT. The participants also stated that due to the high tourist presence in the region, new employees are able to gain experience through early employment before the season.

OJT programs include diverse procedures tailored to firms' sectoral and positional needs, yet there is a notable lack of standardization in implementation. While only 31% of participating firms offer adaptation training, vocational and career development programs, such as communication and reception training, are frequently provided. Positive practices, including incentives, rewards, and additional wages, are implemented by approximately half of the firms to enhance motivation and foster a perception of full-time employment among trainees.

Supervision plays a crucial role in OJT practices. Auditing procedures by TEA were generally viewed positively, with firms commending the constructive supervision style. Yet the apparently smooth coordination between TEA and firms conceals an asymmetry of field power. While TEA formally regulates implementation, employers effectively shape recruitment, supervision, and continuation decisions. This discretionary space, often justified as "flexibility," reproduces dependency on employer goodwill rather than institutional accountability. In this sense, OJT operates as a negotiated order between state and market actors, one that privileges managerial convenience over participant agency. Such dynamics illuminate the institutional fragility underlying otherwise cooperative relationships. Firms reported that audits, focusing on attendance records and compliance, do not create undue pressure, reflecting effective TEA – firm cooperation.

*"They don't come in to punish; they want to see how we manage things. That creates trust." (P18)*

**Table 4. Cluster analysis results of participants' reasons for benefiting from the OJT program.**

| Code A | Code B | Jaccard's Coeffcient |
|---|---|---|
| Social economic impact | Incentives | **0.870968** |
| Improving In-company training for required Qualified Workforce | | 0.741935 |
| Early employment | | 0.225806 |
| Social economic impact | Improving In-company training for required Qualified Workforce | **0.724138** |
| | Early employment | 0.214286 |
| Early employment | Improving In-company training for required Qualified Workforce | 0.2 |
| Experience | Incentives | 0.193548 |
| | Early employment | 0.181818 |
| | Social economic impact | 0.137931 |
| | Improving In-company training for required Qualified Workforce | 0.115385 |

**Note:** This table reflects patterns identified through NVivo-based cluster queries and is intended as a visual supplement to the thematic analysis. Interpretive richness and individual narrative depth remain the primary lens for analysis.

The use of efficient measurement systems allows firms to focus on vocational training. However, challenges remain, such as the dismissal of employees at the program's conclusion and insufficient screening of candidates, particularly during peak seasons, resulting in mismatches within occupational codes.

## 4.3. Perceived outcomes and employment transitions

The OJT program is widely regarded as beneficial, with respondents generally expressing positive views on trainee performance (90%). For instance:

*"Some of our best current employees started through OJT. They were motivated to learn and proved themselves."* *(C25)*

Negative impressions from four firms highlighted issues such as inadequate qualifications, reluctance to learn, and adaptation challenges. Notably, several firms (19 firms) stated that OJT participants outperformed full-time hires in both work ethic and adaptability, though this was not universal. In contrast, others (8 firms) noted lower commitment among trainees, often due to the temporary nature of their contracts:

*"They know it's temporary, so they don't invest in the job. There's a sense of detachment." (C09)*

Businesses benefiting from the OJT program are obliged by law to employ some of the trainees in the program on a full time basis after the training employment period. The most important factors influencing permanent employment decisions were soft skills, not technical performance alone. Participants state that they can continue full-time with OJT trainees who have work discipline, willingness, and determination to learn, career goals and planning, care about personal care, and reliability, especially harmony between employees.

*"We don't just look at how they work; we look at how they behave, how they treat customers and coworkers. That's what counts for us." (C19)*

To better understand firms' preferences in hiring OJT trainees, the cluster analysis function of QDA software was used. NVIVO's cluster analysis here quantifies the co-occurrence of traits like discipline, motivation, and career planning among participants' responses. Fig 4 visualizes how firms group and prioritize soft skill expectations for OJT trainees based on interview data. Cohesion, discipline, career goals, and personal care are the most emphasized criteria for OJT trainees. Perseverance and discipline are often valued together, as are cohesion and personal care. Career planning is highlighted as an independent criterion. Beyond the immediate assessments of performance and adaptability, employers' narratives reveal the structural segmentation of the labor market. Firms' preference for short term incentives and readily adaptable workers reflects the dominance of secondary labor market dynamics, where employment relationships are inherently precarious. Rather than investing in skill deepening, employers often rely on OJT as a mechanism to manage seasonal volatility and labor shortages. This structural logic explains why even positive evaluations of trainees rarely translate into permanent contracts; sustainability becomes constrained not by worker quality but by the economic architecture of the local labor market itself. However, firms face challenges with employees avoiding simple, unqualified jobs, preferring administrative roles, and quitting shortly after being hired. Additionally, candidates recruited through TEA programs often perceive their positions as temporary, which contributes to reluctance and turnover. However, the emphasis on soft skills, discipline, harmony, personal care, also produces unintended consequences. By valorizing behavioral conformity over vocational competence, firms risk reinforcing a moralized understanding of employability. This orientation privileges workers who "fit" existing organizational cultures rather than those who expand their productive capacity. Consequently, OJT may inadvertently perpetuate gendered and classed hierarchies of labor, where personal demeanor outweighs technical

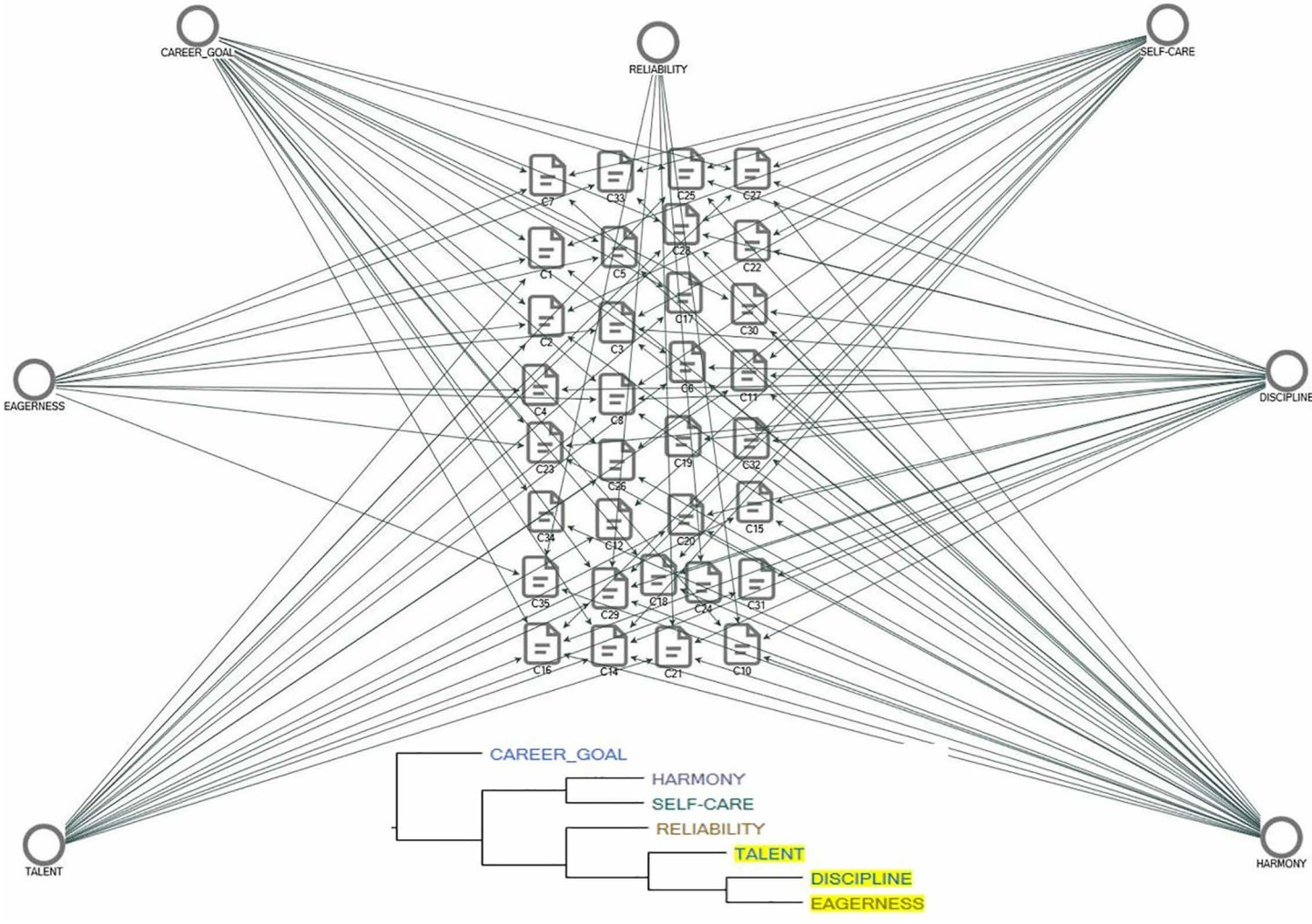

**Fig 4. Cluster analysis of employment criteria after OJT.**

skill in shaping long-term employment opportunities. Despite these challenges, firms view OJT as a significant contributor to reducing unemployment locally and nationally. Many recognize its socio economic impact, with some managers sharing success stories of career advancement following OJT participation, illustrating its potential for long-term benefits.

### 4.4. Criticisms and suggestions for improvement

All participating firms support the program due to the outputs and processes of OJT. However, without sectoral differences, all participants have criticisms and suggestions for the improvement of the OJT program. Their distribution is shown in Fig 5.

The most important criticism of the participating firms on OJT is about Social Security. This refers to the fact that OJT trainees are paid General Health Insurance and Occupational Health and Occupational Disease insurance premiums, but no pension insurance premium is paid. One employer explained that, in practice, TEA covers a portion of the agreed pay while the firm pays the remainder to the trainee; however, informal top-ups and unclear accounting may trigger large retroactive assessments and fines if reported to the Social Security Institution.

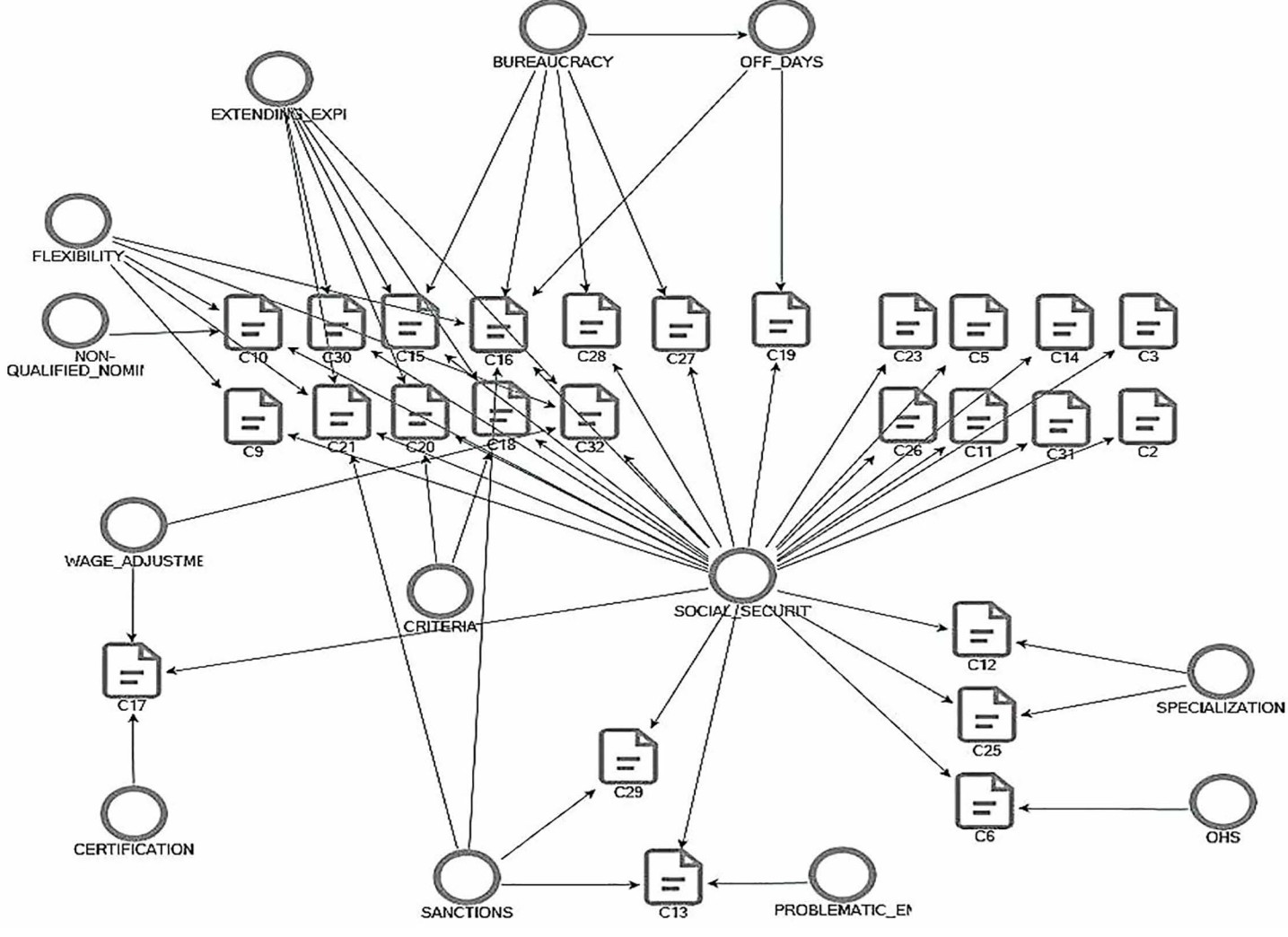

**Fig 5. Participants' Criticisms of OJT.**

To address these issues, firms proposed that OJT trainees be paid pension premiums through the unemployment insurance fund during their program participation. They also criticized the bureaucratic rigidity, particularly regarding fixed working hours, which they find unsuitable for the tourism focused region where the research was conducted. Participants propose a flexible training study program coordinated by TEA to better align with private sector needs and reduce the risk of firms facing sanctions. Additional suggestions include adjusting trainee remuneration based on job types, enhanced prescreening of candidates by TEA considering regional employment criteria, and extending OJT application periods, a recommendation supported by many participants. Despite these concerns, nearly all participants stated they would continue participating in future OJT cycles, especially if policy refinements were made.

Taken together, these patterns suggest that the OJT program's effectiveness cannot be reduced to its training outcomes alone. Employer perspectives uncover the deeper social mechanisms through which labor policy is filtered: economic segmentation, institutional asymmetry, and behavioral expectations. These mechanisms jointly determine whether OJT functions as a vehicle for sustainable employment or as a temporary corrective to structural instability. The

subsequent discussion section builds on this analytical synthesis to conceptualize sustainability within this multi-layered framework.

## 5. Discussion and conclusion

The findings of the study can be meaningfully interpreted through the lens of Human Capital Theory, which posits that investments in education and training enhance workers' productivity and employability, ultimately benefiting both individuals and the broader economy [72]. Specifically, as Crépon and van den Berg (2016) emphasized that the effectiveness of ALMPs depends on program design flexibility and alignment with participants' needs. Employers' critiques, such as bureaucratic rigidity and gaps in social security coverage, reinforce this theoretical argument [2].

Interviews with employers consistently revealed that firms viewed OJT as a tool for training candidates according to firm-specific skill needs, thereby reducing recruitment risk and improving labor quality. Furthermore, the finding that many firms transitioned successful trainees into full-time roles supports the human capital logic that such training yields tangible returns when effectively implemented [73–75]. While OJT can be a powerful human capital investment tool, its effectiveness depends heavily on how it is institutionalized, whom it targets, and which sectors it serves [76]. Although the human capital potential of OJT is significant, programs designed without reference to labor market segmentation are unlikely to address the structural distortions and needs of the labor market. Therefore, policy recommendations should not only focus on skills development but also consider how entrenched patterns of exclusion can be disrupted. The findings are consistent with those of Doerr and Novella's (2020) [61] study conducted in Chile.

Torm (2024) and Hansson (2008) point out that access to and the benefits of OJT are unequally distributed across factors such as age, educational attainment, gender, and sector. They specifically note that low-skilled, older workers, and certain female employees benefit less from OJT programs [24,77]. The findings also indicate that the benefits of OJT are distributed unequally and are largely dependent on sectoral and positional needs. The examined firms tend to perceive OJT participants as occupying the lowest tiers in terms of wages and as part of the unskilled labor force, particularly in wage-setting and career management practices. Although the labor market segmentation approach suggests that for such programs to be effective, different employment conditions should be implemented, wages should be differentiated, and individualized career planning should be provided, current OJT program designs largely lack these elements. The absence of these elements may limit the potential of OJT to produce the qualified personnel needed by the labor market. Moreover, the fact that firms prioritize social skills such as discipline, appearance, and conformity over vocational qualifications reinforces the notion that OJT tends to reproduce, rather than overcome, existing labor market stratification.

The absence of these elements may limit the potential of OJT to produce the qualified personnel needed by the labor market. This segmentation is not only occupational but also demographic. While young people, especially those with little experience, were favored by firms due to their flexibility and perceived trainability, women did not receive any preferential consideration despite their underrepresentation in the Turkish labor market. This finding underscores a critical tension: although OJT programs aim to promote equitable and sustainable employment, the program examined appears to contradict the inequality reduction goals outlined by Jianu et al (2021) [12]. Without the introduction of more inclusive design and monitoring mechanisms in practice, existing forms of exclusion may be reinforced, and the opportunity to strengthen gender equality within an employment process that already involves resource transfers, such as OJT, may be missed. The lack of affirmative action for women, combined with seasonal volatility and rigid program durations, limits the program's transformative potential.

The findings of this study reveal a clear gap between the intended goals of the OJT program and its actual implementation, consistent with the core ideas of the Implementation Gap Perspective. Although the program is designed to improve employability and support sustainable development, problems such as inconsistent candidate screening, lack of standardized training, sectoral mismatches, and the absence of pension insurance limit its real-world effectiveness. These challenges reflect organizational incompatibilities, limited implementation capacity, and weak coordination between TEA and

employers. These factors have also been highlighted by Hudson et al. (2019) and Wamsler and Osberg (2022) [17,36]. The results show that even well-designed policies can fall short if implementation processes are not adapted to local needs and conditions. Addressing these issues requires more flexible and inclusive implementation strategies, along with clearer responsibilities and better communication among stakeholders.

From the perspective of sustainability, the findings reveal that OJT programs in Antalya face significant challenges across the economic, social, and institutional dimensions outlined in the literature. Yet, achieving these dimensions would align OJT directly with SDG 8 – Decent Work and Economic Growth, which emphasizes sustained, inclusive, and productive employment. Economically, firms' dependence on wage subsidies and their limited capacity to retain trainees beyond the incentive period undermine the long-term viability of OJT as a pathway to stable employment, confirming that productivity gains remain short lived. Socially, while the programs offer initial access to employment for women and youth, the absence of systematic mentoring and limited emphasis on capability building restrict their potential to foster inclusive and equitable work outcomes. Institutionally, the fragmented coordination between the Turkish Employment Agency and firms, along with the lack of adaptive mechanisms, constrains the development of enduring public–private partnerships. These results substantiate the argument that sustainability in OJT should be conceptualized not merely as program continuity but as the creation of resilient employment structures, inclusive participation, and institutional learning over time, a perspective consistent with sustainable employability frameworks [15] and the need for embedded, adaptive training systems [16,17].

Among the findings on the structural challenges of OJT programs is the observation that certain employer concerns, such as the pursuit of administrative flexibility, the lack of social security coverage for workers upon retirement, and the emphasis on social skills over formal qualifications, are particularly characteristic of seasonal or tourism-driven economies. While the OJT program aligns rhetorically with SDG 8 by promoting decent work and economic growth, the findings suggest a significant implementation gap. The lack of pension insurance, rigid bureaucratic processes, and the seasonal, short-term nature of employment hinder the realization of "decent" work. Furthermore, the absence of gender-sensitive incentives contradicts SDG 5 on gender equality and weakens the potential of OJT programs to contribute to inclusive labor market outcomes. In the data, this limited inclusivity is reflected in employers' narrow perception of "suitability," where hiring decisions privilege personal discipline, appearance, and social conformity over vocational competence. Such preferences reproduce existing gender and class biases within the labor market rather than dismantling them, indicating that inclusivity is constrained not only by policy design but also by workplace culture and recruitment practices. This highlights the importance of revisiting not only the goals but also the mechanisms through which ALMPs are implemented if they are to genuinely support sustainable development.

Firms generally have a positive outlook on post OJT employment, recognizing its contribution to workforce development. While all firms wish to continue the program, they emphasize the need for adjustments, including reducing bureaucratic rigidity, addressing criminal sanctions, and increasing program flexibility to account for occupational and regional differences in wages and work profiles. The most significant criticism concerns Social Security, specifically the lack of pension insurance premiums for OJT trainees, despite coverage for health and occupational risks. Although OJT entails financial obligations for firms, its expansion and success rely on effective budget utilization and firms' willingness to share responsibility.

Although the empirical findings derive from Antalya, a tourism dominated regional economy characterized by pronounced seasonality and revenue volatility, the analytical insights hold broader relevance for other service oriented labor markets in Türkiye. The logic of analytic generalization Yin, (2014) [78] suggests that qualitative case studies do not aim for statistical representativeness but for conceptual transferability across contexts with comparable institutional conditions. Accordingly, the patterns observed in employers' perceptions of OJT, such as their reliance on short-term labor subsidies, emphasis on behavioral conformity, and challenges in retaining trained staff, are most transferable to regions where employment is heavily influenced by tourism and service fluctuations. In contrast, industrial or agricultural provinces with more standardized occupational hierarchies, stronger unionization, or stable production cycles may exhibit different

dynamics regarding training sustainability and employer engagement. This contextual specificity should therefore guide the cautious interpretation and application of the findings beyond Antalya.

In line with these results, several recommendations are proposed to increase the contribution of TEA's OJT programs to sustainable development and to achieve employment targets more effectively. First, applying proportional positive discrimination for women and young unemployed individuals within the scope of OJT will increase the chances of these disadvantaged groups being employed through the program. Moreover, increasing the incentives offered to firms based on the employment rate achieved at the end of the program will increase employers' motivation and contribute to the organization of new OJT programs. However, supporting the program with differentiated structures and content, such as career planning, wage differentiation, and the enhancement of vocational skills, which align with the various dimensions of labor market segmentation, would be effective in addressing many of the structural challenges identified. Additionally, providing sector specific support to participating firms and promoting employment beyond seasonal periods into longer term work arrangements could yield positive outcomes. Given the view that OJT programs should not be implemented uniformly across all sectors and positions but rather targeted to sectors and roles with specific needs, prioritizing firms producing high-tech and high value-added products—particularly those identified as priorities in national development plans—would enhance the cost–benefit effectiveness of resource allocation.

Future research should center intersectionality as a core analytical framework to more deeply and inclusively assess the impact of OJT programs. The intersection of various socio-demographic positions, such as gender, age, ethnicity, disability status, migration background, and caregiving responsibilities, is crucial for understanding the inclusiveness of these programs. Accordingly, studies should move beyond employer perspectives to focus on the lived experiences of OJT participants, examining the forms of discrimination, exclusion, or empowerment they encounter throughout the program. Furthermore, the impact of OJT should be comparatively analyzed across different sectors (e.g., tourism, industry, agriculture) and geographic contexts (rural/urban), to explore how gender roles and class positions are reproduced within sector specific labor dynamics. Rather than viewing OJT programs solely as a tool for enhancing employability, future analyses should investigate their role in either promoting gender equality or reproducing existing inequalities. This includes developing more equitable and accessible models that account for caregiving responsibilities and demands for flexible work arrangements. In this regard, future research should prioritize questions around how inclusive monitoring and evaluation mechanisms can be designed, and how invisible barriers and implicit forms of discrimination can be identified and dismantled.

## Supporting information

**S1 File. DATASET_INTERVIEWS eng.** English interview dataset including anonymised participant responses used in the analysis.
(XLSX)

## Acknowledgments

The data used in this study is an extended version of the data from the master's thesis titled "İşbaşı Eğitim Programının Etki Değerlendirmesi: Antalya İli Örneği" written by, Zihni Onur EREN under the supervision of Mete Kaan NAMAL, at Akdeniz University, Institute of Social Sciences. New analyses were conducted on the data that had not been used in any previous publications or research.

## Author contributions

**Conceptualization:** Mustafa Koçancı, Mete Kaan Namal.

**Data curation:** Mustafa Koçancı, Mete Kaan Namal, Ufuk Bingöl.

**Formal analysis:** Ufuk Bingöl.

**Investigation:** Zihni Onur Eren.

**Methodology:** Mustafa Koçancı, Mete Kaan Namal.

**Project administration:** Mete Kaan Namal.

**Resources:** Mustafa Koçancı, Mete Kaan Namal, Ufuk Bingöl.

**Software:** Ufuk Bingöl.

**Supervision:** Beyhan Aksoy.

**Validation:** Mustafa Koçancı, Ufuk Bingöl.

**Visualization:** Beyhan Aksoy.

**Writing – original draft:** Mustafa Koçancı, Beyhan Aksoy.

**Writing – review & editing:** Mustafa Koçancı, Beyhan Aksoy.

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
