## [Decision Letter · Decision Letter 0]

7 May 2025

PONE-D-25-06215Is Türkiye's on-the-job training program sustainable? A qualitative evaluation from the firms' perspectivePLOS ONE?

Dear Dr. AKSOY,

Thank you for submitting your manuscript to PLOS ONE. After careful consideration, we feel that it has merit but does not fully meet PLOS ONE’s publication criteria as it currently stands. Therefore, we invite you to submit a revised version of the manuscript that addresses the points raised during the review process.

We look forward to receiving your revised manuscript.

Kind regards,

Turki Talal Turki, Ph.D.

Academic Editor

PLOS ONE

**Journal Requirements:**

1. When submitting your revision, we need you to address these additional requirements. Please ensure that your manuscript meets PLOS ONE's style requirements, including those for file naming. The PLOS ONE style templates can be found at https://journals.plos.org/plosone/s/file?id=wjVg/PLOSOne_formatting_sample_main_body.pdf and https://journals.plos.org/plosone/s/file?id=ba62/PLOSOne_formatting_sample_title_authors_affiliations.pdf 2. We note that you have indicated that there are restrictions to data sharing for this study. For studies involving human research participant data or other sensitive data, we encourage authors to share de-identified or anonymized data. However, when data cannot be publicly shared for ethical reasons, we allow authors to make their data sets available upon request. For information on unacceptable data access restrictions, please see http://journals.plos.org/plosone/s/data-availability#loc-unacceptable-data-access-restrictions.  Before we proceed with your manuscript, please address the following prompts: a) If there are ethical or legal restrictions on sharing a de-identified data set, please explain them in detail (e.g., data contain potentially identifying or sensitive patient information, data are owned by a third-party organization, etc.) and who has imposed them (e.g., a Research Ethics Committee or Institutional Review Board, etc.). Please also provide contact information for a data access committee, ethics committee, or other institutional body to which data requests may be sent. b) If there are no restrictions, please upload the minimal anonymized data set necessary to replicate your study findings to a stable, public repository and provide us with the relevant URLs, DOIs, or accession numbers. Please see http://www.bmj.com/content/340/bmj.c181.long for guidelines on how to de-identify and prepare clinical data for publication. For a list of recommended repositories, please see https://journals.plos.org/plosone/s/recommended-repositories. You also have the option of uploading the data as Supporting Information files, but we would recommend depositing data directly to a data repository if possible. Please update your Data Availability statement in the submission form accordingly. 3. We note that this data set consists of interview transcripts. Can you please confirm that all participants gave consent for interview transcript to be published? If they DID provide consent for these transcripts to be published, please also confirm that the transcripts do not contain any potentially identifying information (or let us know if the participants consented to having their personal details published and made publicly available). We consider the following details to be identifying information:- Names, nicknames, and initials- Age more specific than round numbers- GPS coordinates, physical addresses, IP addresses, email addresses- Information in small sample sizes (e.g. 40 students from X class in X year at X university)- Specific dates (e.g. visit dates, interview dates)- ID numbers Or, if the participants DID NOT provide consent for these transcripts to be published:- Provide a de-identified version of the data or excerpts of interview responses- Provide information regarding how these transcripts can be accessed by researchers who meet the criteria for access to confidential data, including:a) the grounds for restrictionb) the name of the ethics committee, Institutional Review Board, or third-party organization that is imposing sharing restrictions on the datac) a non-author, institutional point of contact that is able to field data access queries, in the interest of maintaining long-term data accessibility.d) Any relevant data set names, URLs, DOIs, etc. that an independent researcher would need in order to request your minimal data set. For further information on sharing data that contains sensitive participant information, please see: https://journals.plos.org/plosone/s/data-availability#loc-human-research-participant-data-and-other-sensitive-data If there are ethical, legal, or third-party restrictions upon your dataset, you must provide all of the following details (https://journals.plos.org/plosone/s/data-availability#loc-acceptable-data-access-restrictions):a) A complete description of the datasetb) The nature of the restrictions upon the data (ethical, legal, or owned by a third party) and the reasoning behind themc) The full name of the body imposing the restrictions upon your dataset (ethics committee, institution, data access committee, etc)d) If the data are owned by a third party, confirmation of whether the authors received any special privileges in accessing the data that other researchers would not havee) Direct, non-author contact information (preferably email) for the body imposing the restrictions upon the data, to which data access requests can be sent 4. Please include captions for your Supporting Information files at the end of your manuscript, and update any in-text citations to match accordingly. Please see our Supporting Information guidelines for more information: http://journals.plos.org/plosone/s/supporting-information.

Reviewers' comments:

Reviewer's Responses to Questions

**Comments to the Author**

1. Is the manuscript technically sound, and do the data support the conclusions?

Reviewer #1: Yes

Reviewer #2: Yes

Reviewer #3: Partly

2. Has the statistical analysis been performed appropriately and rigorously?

Reviewer #1: N/A

Reviewer #2: Yes

Reviewer #3: N/A

3. Have the authors made all data underlying the findings in their manuscript fully available?

Reviewer #1: Yes

Reviewer #2: Yes

Reviewer #3: No

4. Is the manuscript presented in an intelligible fashion and written in standard English?

Reviewer #1: Yes

Reviewer #2: Yes

Reviewer #3: Yes

**Reviewer #1:**  The structure of the subject matter is presented in a coherent manner, effectively highlighting the gaps in evidence and the rationale for the current study.

I believe results shared in this paper are relevant and could inform improvements to the implementation process of the OJT program.

However, there are a few areas that need to be addressed to further improve the quality of evidence presented in this paper.

**Reviewer #2:** A pragmatically important paper indicating that Türkiye's on-the-job training (OJT) program although OJT holds significant potential for promoting sustainable employment, further improvements are necessary to enhance its impact.

**Reviewer #3:**  Dear Authors,

The paper tackles a socially relevant and underexplored topic—employer perspectives on the sustainability and effectiveness of Türkiye's on-the-job training (OJT) programs. Your qualitative analysis adds regional and empirical depth to the literature on active labor market policies.

While the paper is commendable in its scope and empirical richness, I have identified several areas that require substantial revision to enhance clarity, coherence, and scholarly rigor. Please consider the following suggestions:

1. Although the introduction and literature review touch on the relevance of ALMPs and OJT to Sustainable Development Goals, the manuscript lacks a clearly articulated theoretical or conceptual framework guiding the research. Explain your contribution to the literature (e.g., policy evaluation, firm-level engagement with state-sponsored programs, sustainability in active labor market tools) and situate your findings within relevant theoretical debates (e.g., labor economics, human capital theory, institutional theory).

2. The literature review is lengthy and somewhat unfocused. While it provides a broad overview of ALMPs, it could be improved by (a) grouping studies thematically (e.g., the impact of OJT on skills, firm-level perspectives, and gender inclusivity) and (b) highlighting gaps in the existing literature that this study directly addresses. Several citations appear descriptive rather than critically engaged.

3. While Antalya's tourism-dominant labor market is a compelling context, more justification is needed regarding why this province, despite its sectoral specificity, offers generalizable insights about the national OJT program. Are findings region-specific, or do they indicate systemic program features?

4. The research questions are listed clearly but lack integration into the narrative. Each question should link to relevant theoretical expectations or policy debates. Also, consider aligning your findings sections more explicitly with these questions.

5. Using grounded theory and thematic analysis is ambiguous and could confuse readers. Please clarify whether the study adopts an inductive or hybrid coding approach and explain how grounded theory methods (e.g., memo writing, theoretical saturation) were used. Additionally, elaborate on how NVivo was applied beyond coding (e.g., query tools, clustering).

6. The study uses purposive sampling but does not adequately justify the selection of 35 firms or how saturation was assessed. Discuss how sample size and diversity across sectors were sufficient to answer your research questions and whether regional sectoral biases might affect your results.

7. The results section contains rich qualitative insights. However, it is weakened by an over-reliance on figures and tables without adequate textual interpretation. Ensure that each theme is analytically discussed and that quotations are integrated to illustrate key findings. Currently, the results are often read as descriptive summaries.

8. The discussion rightly identifies that gender-based inclusion is lacking in the current OJT system. However, this is not adequately theorized or linked to broader debates in labor policy, equity, or intersectionality. This topic deserves more critical engagement and could serve as a distinctive contribution to the paper.

9. OJT's claim to contribute to sustainable development is repeated but under-analyzed. Sustainability (economic, social, or environmental) should be more clearly defined in the context of labor policy. Consider engaging with SDG 8 ("Decent Work and Economic Growth") more analytically, possibly contrasting policy intentions with implementation realities.

10. The manuscript would benefit from a thorough language edit for clarity, grammar, and flow. Some phrasing is awkward (e.g., "the general approach of firms is harmonious with TEA"), and specific technical terms (e.g., "Jaccard Coefficient" in clustering) are insufficiently explained for a broad audience.

10. While ethics approval is noted, compliance with PLOS ONE data availability standards must be ensured. Currently, stating that data are in Turkish and anonymized is insufficient—clarify the exact conditions for access and provide at least illustrative (translated) data excerpts if permitted.

11. The conclusion reiterates earlier points without offering strong final insights. Please summarize the key contributions, revisit your research questions, and outline more sharply the implications for policy and future research. The recommendations for TEA would benefit from prioritization and contextual grounding.

The manuscript has substantial potential to contribute meaningfully to policy-relevant scholarship on labor markets and sustainability in emerging economies. With a more focused theoretical framework, refined analysis, and improved structure, the article can meet the expectations of PLOS ONE readers.

I encourage the authors to undertake a major revision.

**Do you want your identity to be public for this peer review?** For information about this choice, including consent withdrawal, please see our Privacy Policy

Reviewer #1: No

Reviewer #2: No

Reviewer #3: No

---

## [Author Response · Author response to Decision Letter 1]

6 Jun 2025

Dear Reviewers,

We sincerely appreciate your thoughtful comments and constructive feedback, which have significantly enriched our manuscript. In response, we carefully considered each point raised and revised the manuscript accordingly.

All modifications made following the initial submission are documented in detail in the tracked-changes version of the manuscript. In this version, we have included comment boxes throughout the text specifying exactly which reviewer’s feedback prompted each revision.

Below, we provide a summary of the changes made and our responses to your comments.

RESPONSE TO REVIEWER #1

Reviewer 1 Comment 1:

The decision to only explore managers’ perceptions may provide a one-sided view. Interviewing both managers and employees would improve accuracy. A justification for focusing only on employers may suffice.

Thank you for this insightful comment. In the revised Introduction, we have now provided a clear rationale for focusing exclusively on employer perspectives. We emphasized that firms are the primary agents responsible for implementing OJT practices, making hiring decisions, and evaluating performance—thus offering critical insights into the program's operational effectiveness. While we acknowledge the importance of employee perspectives, we clarify that this will be explored in a planned second-phase study. We have interpreted the data obtained from employers within the framework of human capital, labor market segmentation and implemenatation gap approximations.

Reviewer 1 Comment 2:

Research questions are too descriptive ("what") and suggest a quantitative design. Consider more interpretive phrasing.

We have reworded all four research questions to reflect an exploratory qualitative orientation. For example, the first question now reads: “How do firms recruit participants for the OJT program, and what criteria and methods do they use?” The changes aim to align the inquiry more closely with thematic and interpretive analysis.

We have revised the structure of the Results section to ensure that each thematic finding clearly corresponds to a specific research question. We believe that this alignment enhances the narrative coherence and analytical depth of the manuscript.

Reviewer 1 Comment 3:

Remove the last paragraph that outlines how the paper is structured. (For me, I don't think it is relevant to add the last paragraph on how the paper is sectioned.)

As suggested, the paragraph outlining the manuscript’s structure at the end of the Introduction has been removed to improve flow.

Reviewer 1 Comment 4:

Literature review thoroughly captures up-to-date evidence and what is out there and how it works in this context, with well supported arguments.

Thank you for your feedback. In response to Reviewer 3’s criticisms regarding the literature review, we have engaged in a more theoretical discussion without departing from the content you appreciated. We also aimed for a more systematic presentation by introducing subheadings. Compared to the previous version, we have better reflected the explanatory power of the theoretical discussion and the selected theories in relation to the research problem. We hope that you will find this version more satisfying.

Reviewer 1 Comment 5:

The coding and thematic analysis process is too generic. Please clarify how codes were developed and how themes emerged. (The data analysis section lacks sufficient detail regarding the generation of codes and the determination of final codes and subcodes. Currently, it is overly generic and cannot be replicated. Despite potential word count constraints, it is important to report the process clearly and accurately. The coding process needs to be explained comprehensively to show how codes were derived from the text or how patterns and overarching ideas emerged from combined codes, leading to the identification of themes. Additionally, it should be clarified whether you reviewed the data to confirm that emerging themes accurately represented the codes and original data, and if refinements were needed. Providing these details is crucial to demonstrate the rigour of the analytical process and support the validity of the findings.)

We have substantially revised the materials and methods section. The revised text explains our hybrid approach combining open coding and grounded theory, codebook development, axial theme construction, and validation through inter-coder reliability and NVivo analytics. A clear description of memoing, theoretical saturation, and theme refinement is now provided. We also made an addition to the conclusion section.

Reviewer 1 Comment 6:

Table 3’s use of cluster analysis seems inappropriate for a narrative-based study. Consider removing or reframing. While I appreciate the idea of using a cluster analysis in inferring the benefits of the programme, I wasn't convinced it was appropriate. It is true that cluster analysis groups data based on similarities, often used to identify patterns across datasets – which could apply to qualitative data. However, its utility in interpretive/narrative studies is often limited. This limitation arises because cluster analysis may overlook the intricate, individualised narratives that this study aims to capture. With studies exploring the perceptions of people it is important to maintain the richness and integrity of people's personal experiences, hence why relying on thematic coding and direct interpretation is often better suited. So, to quantify these interpretations, for me, takes away from those experiences. But this is just my opinion. I'd be keen to see what other reviewers think of this.

We greatly appreciate your insights on the methodological alignment between cluster analysis and narrative-based qualitative studies. In light of your comment, we have carefully reconsidered the appropriateness of including cluster analysis in this context. Rather than removing Table 3, we have now reframed its use in the Results section as a supplementary pattern-identification tool, while clearly stating its limitations. We also added a statement in the Discussion noting that such visualizations should not replace interpretive analysis and that future studies might reconsider their use. A limitation regarding this issue is now explicitly included in the new “Limitations” paragraph.

RESPONSE TO REVIEWER #2

A pragmatically important paper indicating that Türkiye's on-the-job training (OJT) program although OJT holds significant potential for promoting sustainable employment, further improvements are necessary to enhance its impact.

Thank you for your encouraging feedback, which motivates us to further improve our research

RESPONSE TO REVIEWER #3

Reviewer 3 Comment 1:

The manuscript lacks a clearly articulated theoretical or conceptual framework.

Although the introduction and literature review touch on the relevance of ALMPs and OJT to Sustainable Development Goals, the manuscript lacks a clearly articulated theoretical or conceptual framework guiding the research. Explain your contribution to the literature (e.g., policy evaluation, firm-level engagement with state-sponsored programs, sustainability in active labor market tools) and situate your findings within relevant theoretical debates (e.g., labor economics, human capital theory, institutional theory).

Thank you for your valuable comment. In the revised manuscript, we have strengthened our analysis by drawing on Human Capital Theory to explain how OJT programs aim to enhance employability by improving workers’ skills through firm-based training. Using Labor Market Segmentation Theory, we highlighted how the benefits of OJT may be unevenly distributed across sectors, occupations, and demographic groups, potentially reproducing existing labor market inequalities. We also employed the Implementation Gap Approach to interpret the disconnection between the policy’s stated objectives and its real-world outcomes. In response, we revised the Introduction section to clearly articulate the study’s conceptual framework. Additionally, we clarified our contribution to the literature in terms of regional implementation dynamics and the role of OJT in supporting SDG 8 (Decent Work and Economic Growth). We are grateful for this suggestion, which has helped us significantly improve the theoretical clarity and scholarly positioning of the manuscript.

Reviewer 3 Comment 2:

The literature review is lengthy and somewhat unfocused. While it provides a broad overview of ALMPs, it could be improved by (a) grouping studies thematically (e.g., the impact of OJT on skills, firm-level perspectives, and gender inclusivity) and (b) highlighting gaps in the existing literature that this study directly addresses. Several citations appear descriptive rather than critically engaged.

In response, we have fully revised the literature review section by thematically organizing it into 3 subsections: (1) OJT and Skill Development, (2) Firm-Level Implementation and Experiences, (3) Gender, Inclusivity, and Sustainability. This structure clarifies the flow of arguments, improves coherence, and helps highlight the specific gaps our study addresses. We also critically engaged with key references to move beyond descriptive summaries and to emphasize the study’s original contribution. Moreover, we have incorporated a clearer articulation of the study's contribution and the gap it fills into the Introduction section.

Reviewer 3 Comment3:

While Antalya's tourism-dominant labor market is a compelling context, more justification is needed regarding why this province, despite its sectoral specificity, offers generalizable insights about the national OJT program. Are findings region-specific, or do they indicate systemic program features?

In response, we revised the introduction and discussion sections to better explain the rationale for selecting Antalya as a case site. We emphasize the region’s sectoral diversity, its prominent role in implementing TEA’s OJT program, and its relevance for identifying systemic implementation challenges. We also clarify which findings are generalizable and which may reflect regional or sectoral specificities.

Reviewer 3 Comment 4:

The research questions are listed clearly but lack integration into the narrative. Each question should link to relevant theoretical expectations or policy debates. Also, consider aligning your findings sections more explicitly with these questions.

In the revised manuscript, we have refined the research questions to better reflect an interpretive qualitative approach and directly tied each question to the conceptual framework. This alignment has enhanced the narrative coherence and analytical depth of the manuscript, and we are grateful for the reviewer’s guidance in improving this aspect of our study.

To further align with the research questions, we also updated the Results section headings as follows:

4.1. Recruitment Practices and Challenges (corresponds to RQ1 & RQ2)

4.2. Experiences with OJT Implementation (corresponds to RQ3)

4.3. Perceived Outcomes and Employment Transitions (corresponds to RQ4)

4.4. Criticisms and Suggestions for Improvement (extends RQ4 into future policy and implementation critique)

Reviewer 3 Comment5:

Clarify grounded theory vs. thematic analysis. Explain how NVivo was used. Using grounded theory and thematic analysis is ambiguous and could confuse readers. Please clarify whether the study adopts an inductive or hybrid coding approach and explain how grounded theory methods (e.g., memo writing, theoretical saturation) were used. Additionally, elaborate on how NVivo was applied beyond coding (e.g., query tools, clustering).

We sincerely thank the reviewer for pointing out this important clarification. In response, we have revised the Data Analysis subsection (Section 3.4) to specify that our study employed a hybrid coding approach, integrating both inductive thematic analysis and grounded theory techniques such as memo writing, constant comparison, and theoretical saturation. We have also elaborated on the use of NVivo 14, detailing how it supported not only code management but also advanced features such as matrix queries, word frequency analysis, and cluster visualizations to enhance analytical depth and transparency. These additions have clarified our methodological rigor and analytic process, as suggested.

Reviewer 3 Comment 6:

Justify sample size and sectoral diversity. The study uses purposive sampling but does not adequately justify the selection of 35 firms or how saturation was assessed. Discuss how sample size and diversity across sectors were sufficient to answer your research questions and whether regional sectoral biases might affect your results.

In the revised manuscript, we have clarified our rationale for selecting 35 firms through purposive sampling by emphasizing sectoral representation, the strategic relevance of Antalya as a region with high participation in on-the-job training, and the diversity of firm types, including those operating in the service sector—primarily tourism—as well as in industry and agriculture. We also now explicitly discuss how thematic saturation was assessed through iterative coding and memo writing, following grounded theory procedures. Additionally, we acknowledge potential regional and sectoral limitations in the discussion, noting that findings may reflect Antalya’s tourism-driven context and encouraging comparative studies across different regions in future research.

Reviewer 3 Comment 7:

Results section is overly descriptive and lacks quotations. The results section contains rich qualitative insights. However, it is weakened by an over-reliance on figures and tables without adequate textual interpretation. Ensure that each theme is analytically discussed and that quotations are integrated to illustrate key findings. Currently, the results are often read as descriptive summaries.

In the revised Results section, we have substantially enriched the narrative by providing more analytical commentary under each theme and integrating direct participant quotations to illustrate key findings. We also revised subheadings to align closely with the research questions, ensuring clearer thematic coherence. While tables and figures remain to support transparency, the textual analysis has been significantly deepened to better reflect the qualitative richness of the data, as recommended.

Reviewer 3 Comment 8:

Gender inequity is mentioned but under-theorized. The discussion rightly identifies that gender-based inclusion is lacking in the current OJT system. However, this is not adequately theorized or linked to broader debates in labor policy, equity, or intersectionality. This topic deserves more critical engagement and could serve as a distinctive contribution to the paper.

In response, we have expanded the Discussion section to engage more critically with the issue of gender inclusion. We now link our findings to broader debates in labor policy, equity, and intersectionality, highlighting the structural and cultural barriers that limit women’s access to OJT opportunities. We also propose targeted policy mechanisms—such as proportional affirmative action or sector-specific gender incentives—to address this gap. We are grateful for this suggestion, which has helped us enhance the paper’s contribution to inclusive labor market discourse.

Reviewer 3 Comment 9:

OJT's claim to contribute to sustainable development is repeated but under-analyzed. Sustainability (economic, social, or environmental) should be more clearly defined in the context of labor policy. Consider engaging with SDG 8 ("Decent Work and Economic Growth") more analytically, possibly contrasting policy intentions with implementation realities.

We expanded our analysis by theorizing sustainability within the framework of SDG 8, and critically examined the gap between the OJT program’s policy rhetoric and its implementation, highlighting economic, social, and environmental dimensions of sustainability in labor policy. Necessary edits have been added to the literature and discussion section.

Reviewer 3 Comment 10:

The manuscript would benefit from a thorough language edit for clarity, grammar, and flow. Some phrasing is awkward (e.g., "the gen

---

## [Decision Letter · Decision Letter 1]

6 Oct 2025

PONE-D-25-06215R1Is Türkiye's on-the-job training program sustainable? A qualitative evaluation from the firms' perspectivePLOS ONE?

Dear Dr. AKSOY,

Thank you for submitting your manuscript to PLOS ONE. After careful consideration, we feel that it has merit but does not fully meet PLOS ONE’s publication criteria as it currently stands. Therefore, we invite you to submit a revised version of the manuscript that addresses the points raised during the review process.

I have read your revised version carefully, as well as your response to the previous referee comments. I think you have addressed well the previous referees' concerns. However, the third referee still recommended a major revision with suggestions provided. I have gone through the suggestions and believe some re-writing could address well those concerns. As your paper is mainly descriptive, further statistical analysis won't generate additional insights on causality. Therefore, I invite you for a minor revision.

We look forward to receiving your revised manuscript.

Kind regards,

Shihe Fu, Ph.D.

Academic Editor

PLOS ONE

Journal Requirements:

Reviewers' comments:

Reviewer's Responses to Questions

**Comments to the Author**

Reviewer #3: (No Response)

2. Is the manuscript technically sound, and do the data support the conclusions?

Reviewer #3: Partly

3. Has the statistical analysis been performed appropriately and rigorously?

Reviewer #3: N/A

4. Have the authors made all data underlying the findings in their manuscript fully available?

Reviewer #3: No

5. Is the manuscript presented in an intelligible fashion and written in standard English?

Reviewer #3: Yes

Reviewer #3: Your study addresses an important and timely issue, namely, the sustainability of Türkiye’s on-the-job training (OJT) program from the viewpoint of firms, and provides valuable empirical insight into a relatively underexplored dimension of active labor market policies (ALMPs).

Your revision has improved the manuscript in several important ways:

1. The manuscript is now better structured, with clearer transitions between the literature review, methods, results, and discussion. The research questions are now more visible and better integrated into the manuscript’s overall narrative.

2. You have provided further detail regarding your qualitative approach, clarified your hybrid use of grounded theory and thematic analysis, and explained how NVivo-supported coding and clustering were conducted. This clarification is helpful, although some ambiguity remains, as I discuss below.

3. The revised results section is now more robust, with a clear thematic structure and better use of direct quotes.

4. The discussion now engages more directly with the Sustainable Development Goals (especially SDG 8) and reflects on the broader policy implications of your findings, including employer perceptions and gender inclusion in training programs.

That said, several important issues remain, and I encourage you to address these in a further revision:

1. While the revised manuscript references SDGs and includes more policy context, it still lacks a compelling conceptual or theoretical framework. Terms such as “sustainability,” “inclusive labor markets,” or “institutional support” are used descriptively but not analytically. Anchoring the study more clearly in relevant theoretical perspectives, such as institutional theory, human capital theory, or the literature on public-private coordination in ALMPs, would deepen the academic contribution.

2. Although your use of direct quotes has increased, many of the findings are still presented in a primarily descriptive manner. There is an opportunity to move beyond summarizing employer sentiments and instead interpret what these views reveal about structural limitations, power dynamics, or unintended consequences of policy design. For instance, your observations about firms’ preference for short-term labor subsidies could be linked to broader discussions on dependency effects or passive employer behavior in ALMPs.

3. The use of cluster analysis via NVivo remains insufficiently justified. If you are maintaining a qualitative paradigm, statistical clustering based on code co-occurrence should be used cautiously and interpreted transparently. Otherwise, it risks appearing as a procedural rather than conceptual contribution. If you choose to retain this technique, please clarify what analytical insight it generated that would not have emerged from thematic coding alone.

4. The rationale for choosing Antalya as the study site has been strengthened, but the question of generalizability remains open. It would be helpful to more explicitly acknowledge the regional specificity of your findings and reflect on how these might vary in other industrial contexts across Türkiye.

5. While you reference SDGs throughout the manuscript, the discussion of sustainability remains somewhat rhetorical. It would strengthen the paper to offer a more precise articulation of what constitutes “sustainability” in OJT, economic viability for firms, social inclusion, long-term employment outcomes, or institutional stability, and to assess how your findings align with or challenge this framework.

6. The revised manuscript is more readable and coherent, but minor grammatical and stylistic issues persist. A final round of professional language editing is recommended to enhance the clarity and fluency of the text.

7. The data availability statement is improved, and the manuscript now includes more illustrative quotes. While this is appreciated, readers would benefit from a more detailed codebook or appendix summarizing your thematic structure (e.g., themes, sub-themes, sample codes) to support transparency and reproducibility further.

**Do you want your identity to be public for this peer review?** For information about this choice, including consent withdrawal, please see our Privacy Policy

Reviewer #3: No

---

## [Author Response · Author response to Decision Letter 2]

24 Oct 2025

Dear Reviewer and Editor,

We sincerely appreciate your thoughtful comments and constructive feedback, which have greatly contributed to improving the quality of our manuscript. In response, we have carefully addressed each point raised and revised the manuscript accordingly. We are uploading the complete dataset as Supporting Information to ensure transparency and facilitate further verification of the study’s findings. All modifications have been clearly indicated in the tracked-changes version of the manuscript. Below, we provide a concise summary of the revisions made and our responses to your comments.

1. In the revised manuscript, we have strengthened the conceptual foundation of the study. While other frameworks, such as institutional theory, were also considered, we found that human capital theory, the implementation gap perspective, and labor market segmentation theory were more suitable for our research focus. Specifically, human capital theory provides a clearer lens for understanding skill formation and productivity outcomes; the implementation gap perspective helps identify discrepancies between policy design and practice; and labor market segmentation theory explains the structural inequalities that constrain the program’s inclusiveness. These approaches were therefore integrated into the analysis to more effectively interpret the dynamics of OJT programs.

Additionally, we have used key terms such as “sustainability” and “inclusive labor markets” in a more analytical and theory-informed manner, both in the literature review and in the discussion of empirical findings. These revisions aim to clarify the study’s theoretical orientation and enhance its overall academic contribution.

2. We enriched the Results section with interpretive paragraphs addressing structural segmentation, institutional power dynamics, and unintended policy consequences. These additions move the findings beyond descriptive summaries by linking employer narratives to systemic mechanisms shaping OJT outcomes. A new closing paragraph now synthesizes these insights and leads into the Discussion section.

3. We thank the reviewer for this critical point regarding the justification of cluster analysis. In response, we have revised Section 3.2 to clarify the methodological role of this technique as an exploratory and complementary tool. The updated text now emphasizes how the analysis visually confirmed the coherence of key thematic clusters, such as discipline–motivation and cohesion–personal care. Furthermore, we have elaborated on how this technique provided a unique analytical insight by demonstrating that the conceptual links we identified were also grounded in the participants' frequently repeated phrases and specific linguistic patterns. This structural evidence reinforces our interpretive findings from a different analytical angle. To enhance transparency, a brief summary of the procedure and the resulting dendrogram have also been included.

4. We added a new paragraph in the Discussion section explicitly addressing analytic generalization and contextual transferability. The revised text explains how findings from Antalya’s tourism-driven economy may apply to other service-oriented regions while outlining the limitations of transferring results to industrial or agricultural contexts.

5. We strengthened the discussion of sustainability by adding a new paragraph in Discussion. It redefines sustainability as a multidimensional construct—economic, social, and institutional—links it explicitly to SDG 8, and ties the concept back to our empirical findings from Antalya, explaining both its current fragility and its potential for long-term transformation.

6. Minor grammatical and stylistic revisions have been made throughout the manuscript. Consistent terminology (e.g., “OJT programs”), uniform table titles, and improvements in sentence fluency and clarity have been carefully implemented.

7. We would like to thank the reviewer for their insightful and constructive feedback. We appreciate their acknowledgment of the improved data availability statement and the inclusion of additional illustrative quotes. In response to the valuable suggestion for enhancing the transparency and reproducibility of our research, we have now supplied a comprehensive codebook, a thematic file detailing the hierarchy of our themes, categories, and codes, and the fully anonymized raw data as supplementary materials with this revision. We are confident that these additions will provide readers with a clear and thorough overview of our analytical process, directly addressing the reviewer's comments and further strengthening the manuscript.

---

## [Decision Letter · Decision Letter 2]

2 Jan 2026

PONE-D-25-06215R2Is Türkiye's on-the-job training program sustainable? A qualitative evaluation from the firms' perspectivePLOS One?

Dear Dr. AKSOY,

Thank you for submitting your manuscript to PLOS ONE. After careful consideration, we feel that it has merit but does not fully meet PLOS ONE’s publication criteria as it currently stands. Therefore, we invite you to submit a revised version of the manuscript that addresses the points raised during the review process.

We look forward to receiving your revised manuscript.

Kind regards,

Md Jahangir Alam, PhD

Academic Editor

PLOS One

Journal Requirements:

Additional Editor Comments:

This paper does not need further review; rather, it will be accepted based on this minor revision.

Reviewers' comments:

Reviewer's Responses to Questions

**Comments to the Author**

Reviewer #4: (No Response)

2. Is the manuscript technically sound, and do the data support the conclusions?

Reviewer #4: Yes

3. Has the statistical analysis been performed appropriately and rigorously?

Reviewer #4: Yes

4. Have the authors made all data underlying the findings in their manuscript fully available?

Reviewer #4: Yes

5. Is the manuscript presented in an intelligible fashion and written in standard English?

Reviewer #4: Yes

Reviewer #4: The manuscript is well written but I have few suggestions to be addressed:

1. The authors should use academic language avoiding personal pronoun e.g., used "We" in line no 383 on page 23 and many more.

2.. The research questions should be come after literature review. Before research questions put a paragraph mentioning the research gaps that this study going to fill up.

**Do you want your identity to be public for this peer review?** For information about this choice, including consent withdrawal, please see our Privacy Policy

Reviewer #4: No

---

## [Author Response · Author response to Decision Letter 3]

2 Jan 2026

Response to Reviewer

We would like to thank the Academic Editor and the reviewer for their careful evaluation of our manuscript and for the constructive comments that helped improve its clarity and structure. All comments have been fully addressed, as detailed below.

Reviewer

Comment 1:

The authors should use academic language avoiding personal pronouns (e.g., “we”).

Response:

The manuscript has been thoroughly revised to remove all first-person plural pronouns (e.g., “we,” “our,” “us”) from the academic narration. An impersonal academic style has been adopted consistently throughout the text, while preserving the original meaning, analytical arguments, and methodological rigor of the study.

Comment 2:

The research questions should be placed after the literature review. Before the research questions, a paragraph should be included to clarify the research gap addressed by the study.

Response:

The Research Questions have been reorganized to follow the Literature Review, placed immediately after the paragraph at the end of Section 2.3 that identifies the research gap and outlines the contribution of the present study. This reorganization establishes a smooth and coherent transition from the literature review to the methodological section by explicitly linking the identified gaps to the research questions that guide the empirical analysis.

---

## [Editor Report · Decision Letter 3]

5 Jan 2026

Is Türkiye's on-the-job training program sustainable? A qualitative evaluation from the firms' perspective

PONE-D-25-06215R3

Dear Dr. AKSOY,

We’re pleased to inform you that your manuscript has been judged scientifically suitable for publication and will be formally accepted for publication once it meets all outstanding technical requirements.

Kind regards,

Md Jahangir Alam, PhD

Academic Editor

PLOS One
---

## [Editor Report · Acceptance letter]

PONE-D-25-06215R3

PLOS One

Dear Dr. AKSOY,

I'm pleased to inform you that your manuscript has been deemed suitable for publication in PLOS One. Congratulations! Your manuscript is now being handed over to our production team.

Kind regards,

on behalf of

Dr. Md Jahangir Alam

Academic Editor

PLOS One